# A New Biofertilizer Formulation with Enriched Nutrients Content from Wasted Algal Biomass Extracts Incorporated in Biogenic Powders

Fran Nekvapil [1,2,3], Iolanda-Veronica Ganea [2,4], Alexandra Ciorîță [5,6], Razvan Hirian [7], Sanja Tomšić [8], Ildiko Melinda Martonos [4] and Simona Cintă Pinzaru [1,3,*]

1    Ioan Ursu Institute, Babeş-Bolyai University, 400084 Cluj-Napoca, Romania; fran.nekvapil@ubbcluj.ro
2    Physics of Nanostructured Systems Department, National Institute for Research and Development of Isotopic and Molecular Technologies, 400293 Cluj-Napoca, Romania; iolanda.ganea@itim-cj.ro
3    RDI Laboratory of Applied Raman Spectroscopy, RDI Institute of Applied Natural Sciences (IRDI-ANS), Babeş-Bolyai University, 400293 Cluj-Napoca, Romania
4    Faculty of Environmental Science and Engineering, Babes-Bolyai University, 400294 Cluj-Napoca, Romania; ildiko.martonos@ubbcluj.ro
5    Electron Microscopy Laboratory, Faculty of Biology and Geology, Babes-Bolyai University, 400015 Cluj-Napoca, Romania; alexandra.ciorita@itim-cj.ro
6    Integrated Electron Microscopy Laboratory, National Institute for Research and Development of Isotopic and Molecular Technologies, 400293 Cluj-Napoca, Romania
7    Department of Condensed Matter Physics and Advanced Technologies, Babeş-Bolyai University, 400084 Cluj-Napoca, Romania; razvan.hirian@ubbcluj.ro
8    Department of Applied Marine Ecology, University of Dubrovnik, 20000 Dubrovnik, Croatia; sanja.tomsic@unidu.hr
*    Correspondence: simona.pinzaru@ubbcluj.ro; Tel.: +40-264-405-300-5188; Fax: +40-264-591-906

**Abstract:** Raw algae waste naturally thrown on shores could pose serious threats for landfilling and its reuse for composting or further processing as added-value by-products require knowledge-based decisions and management for the sustainable development of local ecosystems. Raw marine waste containing salt (halite) and heavy metals that eventually accumulate in algae hamper their safe applicability in soil fertilization or in other exploitations. Here, the suitability of algal biomass for use as an environmentally safe fertilizer was investigated, thereby supporting sustainable coastal management. The simple extraction of the dry algal biomass of three abundant Mediterranean species, *Enteromorpha intestinalis*, *Corallina elongata*, and *Gelidium pulchellum*, in water containing sodium carbonate resulted in a greenish extract containing a reduced heavy metals content, and nutrients such as $K^+$, $PO_4^{3-}$, $SO_4^{2-}$, $NO_3^-$, $Ca^{2+}$, and $Mg^{2+}$. UV-Vis and Raman techniques, including surface-enhanced Raman scattering (SERS), were employed for the fast evidencing of polyphenols, carotenoids, and chlorophylls in the extracts content, while *E. intestinalis* extract additionally exhibited polysaccharide signals. Heavy metals analysis showed that the major metals in the extracts were Fe, Ni, Zn, and Cu; however, their levels were an order of magnitude lower than in the dry biomass. The extracts also showed a mild antibacterial effect. The combination option of aqueous extracts with powdered crustacean shells to obtain a novel, eco-friendly, solid biofertilizer complex was further shown, which could be pelleted for convenient use. The immersion of solid biofertilizer pellets in water is accompanied by re-solubilization of the compounds originating from algae extracts, presenting the opportunity for dry storage and easier handling and land applicability. In summary, aqueous extracts of marine algae waste present an environmentally safe and attractive way to recycle excessive algal biomass and to formulate a new, eco-friendly biofertilizer complex.

**Keywords:** biofertilizer; algal biomass; aqueous extracts; environmental legislation; UV/Vis; surface-enhanced Raman spectroscopy; heavy metals; antibacterial effect

## 1. Introduction

The use of algae biomass for fertilization purposes has been known from the early human civilizations [1]. This proves that our ancestors were aware of their agricultural importance and benefits by observation, and today, we know this stems from enhancing nutrient availability or modifying the soils pH by acting as nitrogen-fixing agents, sources of organic matter, or as deposits of inorganic nutrients [1–3]. Macroalgae biomass is well-known as a significant resource of exploitable products mostly used for valuable compounds extraction, the food industry, or composting for soils fertilization. At sites under local input of nutrients, the rapid growth of algal and seaweed biomass that is washed on shore may impede socio-economic activities in coastal zones, such as tourism, boating or swimming, and recreational activities and may pose serious threats for local ecosystems. Artificial eutrophication (e.g., through coastal run-offs) can cause massive algae blooming with adverse effects on aquatic biota [1] and impediments to other socio-economic activities such as coastal tourism and beach recreation. On the other hand, this algae biomass has a great applicative potential as a cheap source of natural biofertilizers for different cropping systems, allowing the recycling of previously discharged nutrients. The salt, heavy metals, and potential harmful bacterial content of raw algae waste naturally thrown on shores could pose serious threats for landfilling, and their reuse for composting would require long-time handling and management, but the salt (halite) and heavy metals content hamper their safe applicability in soil fertilization.

The rapid increase in global population in the last century has triggered a great environmental pressure and a high demand for high-quality food products. This has led to an expansion in both the surface of cultivable areas and in the number and dosage of agrochemicals applied [4] for the stimulation of richer agricultural products, with resistance to diseases and pests. Although classic chemical fertilizers have many financial advantages, they proved to pose a significant toxicological risk for biota integrity and human health [5]. Therefore, researchers have focused on finding green alternatives that promote ecological and sustainable agricultural practices [6]. In this context, the marine algae waste, having the largest distribution in nature, has been commercialized as crops biostimulants due to their biodegradability and rich content of nitrogen, potassium, phosphorous, calcium, magnesium, etc. [7]. The species *Turbinaria ornata*, *Ulva reticulata*, *Ulva lactuca*, *Brassica napus*, and *Gelidium crinale* are some examples of marine algae used as biofertilizers [8]. Moreover, it was also reported that seaweeds contain bioactive compounds and generate secondary metabolites, which proved effective against several phytopathogens and nematodes [9,10].

Algae can be collected from natural meadow along the shallow coast at eutrophic sites or may also be cultured in nutrient-enriched effluents, such as wastewaters low in hazardous contaminants [11] or effluents from *integrated multi-trophic aquaculture* facilities [12–15], where the daily growth rate has been reported as up to 2.6% [14] for macroalgae *Ulva lactuca* and *Gracilaria chinensis* cultured in abalone effluent, and from 1% to 4% when cultured in mullet farm effluent [13] and *Gracilaria birdiae* cultured in shrimp farm wastewater [15]. Some of the sources of eutrophication are persistent and bioaccumulative, becoming a great concern as they threaten delicate communities and ecosystems. Hence, the supply of algae biomass grown on effluent nutrients or run-offs seems sustainable and feasible.

The present cross-disciplinary study investigated whether the extracts obtained from the three algae species abundant in the southeastern Adriatic Sea during late spring and summer, *Enteromorpha intestinalis*, *Corallina officinalis*, and *Gelidium pulchellum* feature suitable nutrient and biochemical contents for use in plant fertilization. We showed that aqueous extracts have a reduced heavy metals content relative to the starting biomass, and we proposed the combination of liquid extracts with powdered crustacean shells to obtain a novel, pelletable, dried complex biofertilizer, featuring both the mineral component (biocomposite powder from shells) and the soluble nutrient components from algae extract. The fundamental refining steps for raw biomass, including drying, extraction, and

combination with a solid biogenic porous material, are presented to facilitate its handling and environmental safety applicability following the Blue bioeconomy concept.

This study proposed a novel route to utilize excess biomass originating from aquatic vegetation primary production [2,16,17]. This provides decision-making bodies, and beach and waste managers, with a greater palette of solutions to tailor the biomass management according to the local logistics and demands.

## 2. Materials and Methods

### 2.1. Algae Biomass and the Extraction Procedure

Three species of coastal shallow-water macroalgae representing a notable coastal ecosystem component in spring-early summer were considered: (1) *C. officinalis*, (2) *G. pulchellum*, and (3) *E. intestinalis*. The biomass of the three species was collected along the Mediterranean coastal area in Dubrovnik (Croatia), characterized by moderate touristic activity producing sewage. However, this ecosystem still holds an oligotrophic designation. The communities and habitat conditions associated with the site are characterized as biocenosis of the lower mediolittoral rocks and biocenosis of infralittoral algae. These habitats are present and widespread along the mainland and islands of the entire eastern Adriatic coast.

For effective preservation during the transport and storage before analyses, the biomass was shredded and dried in an oven at 60 °C for 72 h. Extraction was performed by boiling the dried algal biomass, which was powdered in an agate mortar, at 100 °C for 100 min in the presence of a 2% solution of $Na_2CO_3$. The aqueous phase/biomass weight ratios were 26:1 and 52:1, respectively, in different preparations. After extraction, the supernatant was filtered, and aliquots were taken for further analysis.

### 2.2. Algae Extracts and Biomass Characterization

The content of dissolved ions ($Na^+$, $K^+$, $Mg^{2+}$, $Ca^{2+}$, $F^-$, $Cl^-$, $NO_3^-$, $PO_4^{3-}$, and $SO_4^{2-}$) in the extracts was analyzed with an ICS 1500 Dionex Ion Chromatography System (Dionex Corporation, Sunnyvale, CA, USA), equipped with a dual-piston serial pump.

The UV-Vis absorption measurement of the algal extracts was conducted on a Jasco V550 UV-Vis spectrometer (Jasco Corporation, Tokyo, Japan), within a range from 200 to 900 nm with a 0.5 nm step. The measurements required a 2-fold dilution of the 26:1 extract with distilled water.

Raman spectroscopy measurements of the extracts were conducted on a Renishaw InVia Reflex confocal Raman microscope (Renishaw, Wotton-under-Edge, UK), configured to analyze bulk samples. All samples were combined with colloidal silver nanoparticles synthetized by the Lee–Meisel method [18] to achieve surface-enhanced Raman scattering (SERS). A Cobolt DPSS laser (Cobolt AB, Solna, Sweden), emitting at 532 nm, was used for excitation.

Samples of 0.4 g of algae dry biomass, and solid residue extracts of each species, were prepared for the heavy metals content analysis by applying a prior microwave-assisted digestion procedure using 5 mL of $HNO_3$ and 5 mL of $H_2O_2$, and, respectively, 5 mL of $HNO_3$ and 3 mL of $H_2O_2$. The microwave programs followed four heating steps with a 2 min ramp: at 145 °C for 5 min, 170 °C for 5 min, 190 °C for 15 min, and 75 °C for 10 min. After cooling, the samples were transferred into volumetric flasks and diluted to a final volume of 25 mL with 0.2% $HNO_3$. Analytical blanks were also prepared using the same procedure. For metal determination, all algae samples (extracts and mineralized samples) were analyzed with a ZEEnit 700 Atomic Absorption Spectrometer (AAS) (Analytik Jena, Jena, Germany) equipped with single-element hollow cathode lamps, an air-acetylene flame, and a graphite furnace.

The antibacterial effect of the algae extracts was assessed according to EUCAST protocols using the microdilution method [19]. Briefly, the algae suspension (100 μL) was inoculated in 96-well plates at a final concentration/well of 4540 μg $mL^{-1}$ and a final volume of 200 μL per well. Serial dilutions were performed, and a range of 10 concen-

trations was obtained (from 8 μg mL$^{-1}$ to 4540 μg mL$^{-1}$), after which 20 μL of bacterial suspension of 0.5 McFarland turbidity was added to each well. Each plate contained one untreated control and one positive control with bacteria treated with ciprofloxacin of 10 μg mL$^{-1}$ concentration. The bacterial strains were American Type Culture Collection (ATCC) standards (ATCC, Manassas, VA, USA): *E. coli* ATCC 25922, *S. aureus* ATCC 25923, *P. aeruginosa* ATCC 27853, and *E. faecalis* ATCC 29212. The plates were left to incubate for 24 h at 35 °C after which the absorbance was read at 600 nm wavelength using the BioTek Epoch plate reader (BioTek Instruments, Winooski, VT, USA).

### 2.3. Adsorption and Release Test Using Crustacean Shell Powder Loaded with Algae Extracts

Crab shell micropowder described by Nekvapil et al. [20] and consisting in large part of nanoporous Mg-CaCO$_3$ [20] was used as a solid biogenic porous carrier material. In brief, crab shells of the species *Callinectes sapidus* (the Atlantic blue crab) and *Carcinus aestuarii* (the Mediterranean green crab) were ground into sub-100 μm particles by planetary ball milling.

Liquid algae aqueous extract of each species was separately loaded onto the shell micropowder by combining an equal volume of the extract and the shells (loading step). This step, which resulted in material of paste-like consistency, was followed by drying overnight at 40 °C in a ventilated oven to obtain a dried complex featuring the benefits of both the biogenic shell c and the algae content, and it was called the biofertilizer complex.

To test the desorption of the compounds extracted from algae, aliquots of 1 g of the biofertilizer complex were compressed into pellets by 3 t of pressure using a hydraulic press. Subsequently, a biofertilizer complex pellet containing extracts of each alga species was re-dispersed in 200 mL of distilled water (release step) on a magnetic stirrer (magnet rotation speed 500 rpm). The subsamples taken from the dispersion 1 h after the addition of respective pellets were combined with AgNPs and analyzed by exploiting the surface-enhanced Raman scattering effect for sensitive detection of organic species.

## 3. Results

### 3.1. Algae Extraction

The algae extracts after filtration were greenish in color and slightly more viscous than water (Figure 1). The extracts featured a low-intensity, seawater-like odor, which was consistent even after one year of preservation under cold and dark conditions.

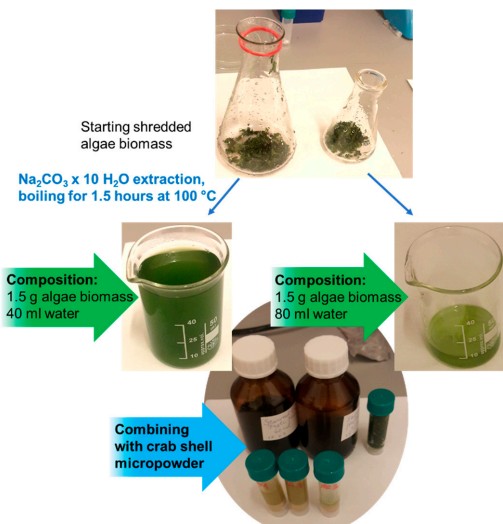

**Figure 1.** Photographic display of the algae extraction procedure and combination with crab shell powder, referring to cases when 1.5 g of dry algae biomass was extracted in either 40 or 80 mL of water.

### 3.2. Algae Extracts and Biomass Characterization

### 3.2.1. Dissolved Cations and Anions Content

The content of dissolved ions in the algae extracts is presented in Table 1. Here, the ratio of the ions should be the most relevant parameter, as their absolute concentration is expected to scale with changes in the extraction water volume. The most abundant cations are $K^+$ and $Ca^{2+}$, while the predominant anions are $Cl^-$, $SO_4^{2-}$, and $NO_3^-$. The content of $Na^+$ originating from algae could not be reliably determined, as it is present both in the residual salt from seawater and in the use of $Na_2CO_3$ as the extraction catalyst. In general, $Cl^-$ along with $Na^+$ are the most prevalent ions in seawater, representing 85% of its total dissolved ions. Likewise, $Ca^{2+}$, $Mg^{2+}$, $K^+$, and $SO_4^{2-}$ can emerge from the same natural source [21]. The $NO_3^-$, $F^-$, and $PO_4^{3-}$ input can be associated with nearby sources of domestic sewage around the collection site. The last two anions can also originate via continental weathering [22]. Comparing the investigated algae species, we note that *C. officinalis* contained greater proportions of $Ca^{2+}$ (5.00 mg $L^{-1}$), $F^-$ (1.89 mg $L^{-1}$), and $PO_4^{3-}$ (0.68 mg $L^{-1}$). Overall, the extract of *G. pulchellum* was rich in $K^+$ (420.00 mg $L^{-1}$) and $NO_3^-$ (6.65 mg $L^{-1}$), and poor in $Ca^{2+}$ (2.80 mg $L^{-1}$) and $PO_4^{3-}$ (0.46 mg $L^{-1}$), reporting no traces of $F^-$. Thus, we can state that the tested algae species had distinct profiles of dissolved ions.

**Table 1.** Content of dissolved ions (mg $L^{-1}$) determined in the aqueous extracts of *Corallina officinalis*, *Gelidium pulchellum*, and *E. sp.*, obtained using a 26:1 water to biomass *w/w* ratio (1.5 g starting biomass).

| * Cations/Anions | *Corallina officinalis* | *Gelidium pulchellum* | *Enteromorpha intestinalis* |
|---|---|---|---|
| $Na^+$ | 6272.1 ± 25.6 | 7018.1 ± 29.1 | 7818.0 ± 34.5 |
| $K^+$ | 157.4 ± 11.1 | 420.0 ± 16.3 | 381.6 ± 14.2 |
| $Ca^{2+}$ | 5.0 ± 0.1 | 2.8 ± 0.0 | 16.4 ± 1.7 |
| $Mg^{2+}$ | 0.1 ± 0.0 | 39.5 ± 3.5 | 128.1 ± 10.2 |
| $Cl^-$ | 33. 7 ± 3.1 | 177.5 ± 13.8 | 514.4 ± 17.6 |
| $SO_4^{2-}$ | 8.8 ± 0.4 | 34.9 ± 2.9 | 55.5 ± 4.9 |
| $NO_3^-$ | 2.9 ± 0.1 | 6.7 ± 0.3 | 0.3 ± 0.0 |
| $F^-$ | 1.9 ± 0.0 | 0.0 ± 0.0 | 0.8 ± 0.0 |
| $PO_4^{3-}$ | 0.7 ± 0.0 | 0.5 ± 0.0 | 0.6 ± 0.0 |

* Values are expressed as mean ± standard deviation (*n* = 3).

### 3.2.2. Electronic Absorption Spectroscopy

UV-Vis absorption spectroscopy is a well-recognized method for revealing the sample pigments [23] and phenolics [24] content, and the spectra acquired from the extracts of the three species concerned are presented in Figure 2. Regarding pigments, the features at 580, 627, and 650 nm can be assigned to levels of different chlorophylls [25]. However, it is curious that *G. pulchellum* extract did not exhibit a chlorophyll signal. The weaker absorption bands at 445, 500, and 535 nm, recorded only in *E. intestinalis* extract, could indicate the presence of carotenoids [26,27]. However, it is unclear to which extent carotenoids and chlorophylls play a role in plant fertilization.

The features in the 260–320 nm range are characteristic of phenolic compounds [28]. The feature around 320 nm, which appears as a defined band in the spectrum of *G. pulchellum*, but only as a shoulder in *C. officinalis,* arises from the phenyl ring functionalization, while the <280 nm features arise from modifications of the aromatic ring itself.

The origin of the bands at 405 and 445 nm is not clear; however, they may arise from the phenolic glucosides under elevated pH [29] (basic extraction conditions). This assumption is plausible because green algae from the Ulvales order are known to be rich in sulphated polysaccharides. Ray [30] proposed that polysaccharides of a related species, *Enteromorpha compressa*, are composed of glucose (glc) and xylose (xyl)-sulphated oligomer building blocks $Glc_3Xyl_1$, $Glc_4$, $Glc_3Xyl_2(SO_3Na)_2$, $Glc_4Xyl_2(SO_3Na)_2$, and $Glc_5Xyl_2(SO_3Na)_2$, determined after sequential inorganic solvents extraction steps. Thus, the spectra indicate

that *C. officinalis* and *G. pulchellum* may be rich in hydroxylated phenols (320 nm feature) [28,31], while *E. intestinalis* extract may be rich in glucoside-containing polyphenols (405 and 445 nm feature).

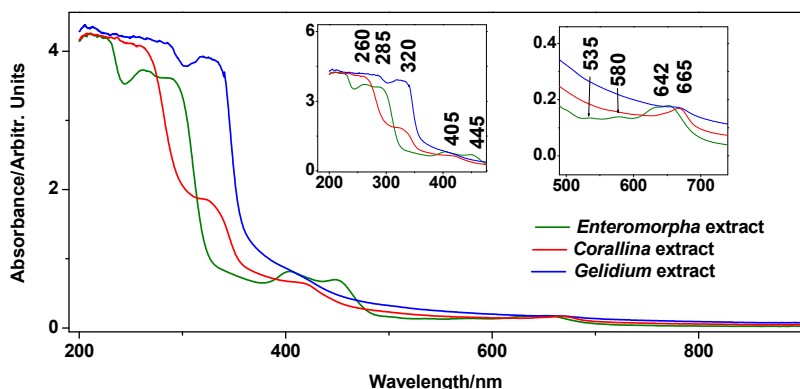

**Figure 2.** UV-Vis absorption spectrum of the *Corallina officinalis*, *Gelidium pulchellum*, and *Enteromorpha intestinalis* aqueous extracts. The insets show the enhancements of ca. 200–500 nm and 500–750 nm ranges.

### 3.2.3. Raman Spectroscopy

Normal Raman spectra of the extracts exhibited a high fluorescence background, hampering any characteristic normal Raman signal, as expected from polyphenols-rich samples; thus, the surface-enhanced Raman scattering (SERS) technique appeared as an attractive alternative for probing their content. For fast tracking of the presence of organic components in extracts, we employed SERS with recognized advantages in the sensitive detection of low-concentration analytes. SERS spectra were recorded using the classical Ag colloidal nanoparticles [18] for Raman enhancement, and a small amount of about 10 μL of extract was added to 500 μL of AgNPs and gently stirred before SERS data collection. The SERS signal of the three tested algae species is presented in Figure 3. In the case of *C. officinalis* and *G. pulchellum*, the features arise from phenolic compounds: the 870–960 cm$^{-1}$ feature, the 1097 cm$^{-1}$ band, the 1440–1480 features, and the band at 1575 cm$^{-1}$ [32]. The proposed band assignments to atomic vibrational modes are given in Table S1. According to the curves presented by Aguilar-Hernandez et al. [32] in their Figure 3, SERS spectra of dissolved phenolic compounds (colloidal solution) feature wide and poorly defined bands, which is similar to the overall weak character of SERS spectra acquired from *C. officinalis* and *G. pulchellum* extracts here.

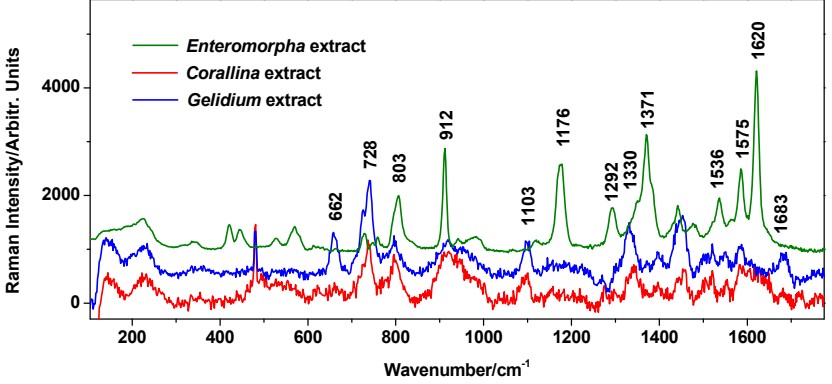

**Figure 3.** SERS spectra (background subtracted) of the *Enteromorpha intestinalis*, *Corallina officinalis*, and *Gelidium pulchellum* extracts. Spectra were acquired under 532 nm.

In the SERS spectrum of *E. intestinalis* extract, the additional medium-intensity band at 1176 cm$^{-1}$ indicates the C-O-C stretch of the glycosidic bond, while the stronger band at

1620 cm$^{-1}$ and the band at 1536 cm$^{-1}$ are assigned to the COO$^-$ carboxylate asymmetric stretching in saccharides [33].

### 3.2.4. Heavy Metals Content

The metal content in the aqueous *C. officinalis*, *G. pulchellum*, and *E. intestinalis* extracts (Table 2) was notably lower than in the content of respective dry biomass (Table S2). We noticed that Fe, Ni, Zn, and Cu were the most abundant metals in the aqueous extracts. *G. pulchellum* extract contained the largest amount of metal ions (except for Fe) and was the only sample in which Mn was detected, while Cr was below the detection limit in all three extracts. The study also revealed that the *E. intestinalis* extract showed the lowest Fe, Zn, Cu, and Cd levels among the three species. The overall low metal content in the three extracts compared to that from raw materials may be a consequence of the extraction procedure applied here.

**Table 2.** Heavy metals content (mg L$^{-1}$) determined in the aqueous extracts of *Corallina officinalis*, *Gelidium pulchellum*, and *Enteromorpha intestinalis*.

| * Metal Ions | *Corallina officinalis* | *Gelidium pulchellum* | *Enteromorpha intestinalis* |
|:---:|:---:|:---:|:---:|
| Fe | 0.296 ± 0.009 | 0.233 ± 0.008 | 0.085 ± 0.003 |
| Ni | 0.111 ± 0.003 | 0.169 ± 0.005 | 0.129 ± 0.004 |
| Zn | 0.095 ± 0.002 | 0.389 ± 0.002 | 0.037 ± 0.001 |
| Cu | 0.091 ± 0.002 | 0.348 ± 0.004 | 0.066 ± 0.001 |
| Pb | 0.053 ± 0.004 | 0.065 ± 0.005 | 0.062 ± 0.004 |
| Cd | 0.038 ± 0.002 | 0.044 ± 0.002 | 0.027 ± 0.001 |
| Mn | 0.000 ± 0.0 | 0.030 ± 0.001 | 0.000 ± 0.0 |
| Cr | 0.000 ± 0.0 | 0.000 ± 0.0 | 0.000 ± 0.0 |

* Values are expressed as mean ± standard deviation ($n$ = 3).

The metal content determined in the digested dry biomass (step before extraction) revealed the native heavy metals content of the investigated algae species (Supplementary Table S2). The native biomass was richer in Fe, Pb, Zn, Cu, Mn, and Ni, while Cd and Cr were present in low concentrations.

Considering the levels of the contaminants in the starting dry biomass and the respective extracts (Figure S2), the highest metals extraction rates associated with *C. officinalis* and *G. pulchellum* were found for Cd (27.512% and 47.476%, respectively), Ni (22.573% and 33.641%, respectively), Cu (12.738% and 12.901%, respectively), and Zn (10.052% and 14.384%, respectively), while the lowest values were for Fe, Mn, and Cr (Figure S1). Concerning the extraction rates obtained for *E. intestinalis*, the highest rates were observed in the cases of Pb (44.035%), Ni (38.583%), and Cd (31.443%).

### 3.3. Microbiology

The microdilution method revealed a dose-dependent response for *E. coli* and *S. aureus*, which were inhibited with increasing biomass concentrations (Figure 4). On the other hand, *E. faecalis* had a proliferative response when treated with *C. officinalis* and *G. puchellum*, and a slight inhibition was achieved with the use of *E. intestinalis* at low concentrations (8 and 17 μg mL$^{-1}$). *P. aeruginosa* was inhibited in an almost constant manner by *E. intestinalis* (values between 80 and 90% inhibition), while the highest concentrations of *C. officinalis* and *G. pulchellum* (4540 μg mL$^{-1}$) had a good inhibitory capacity (values under 50% inhibition). None of the extracts, however, were comparable with the inhibitory capacity of ciprofloxacin.

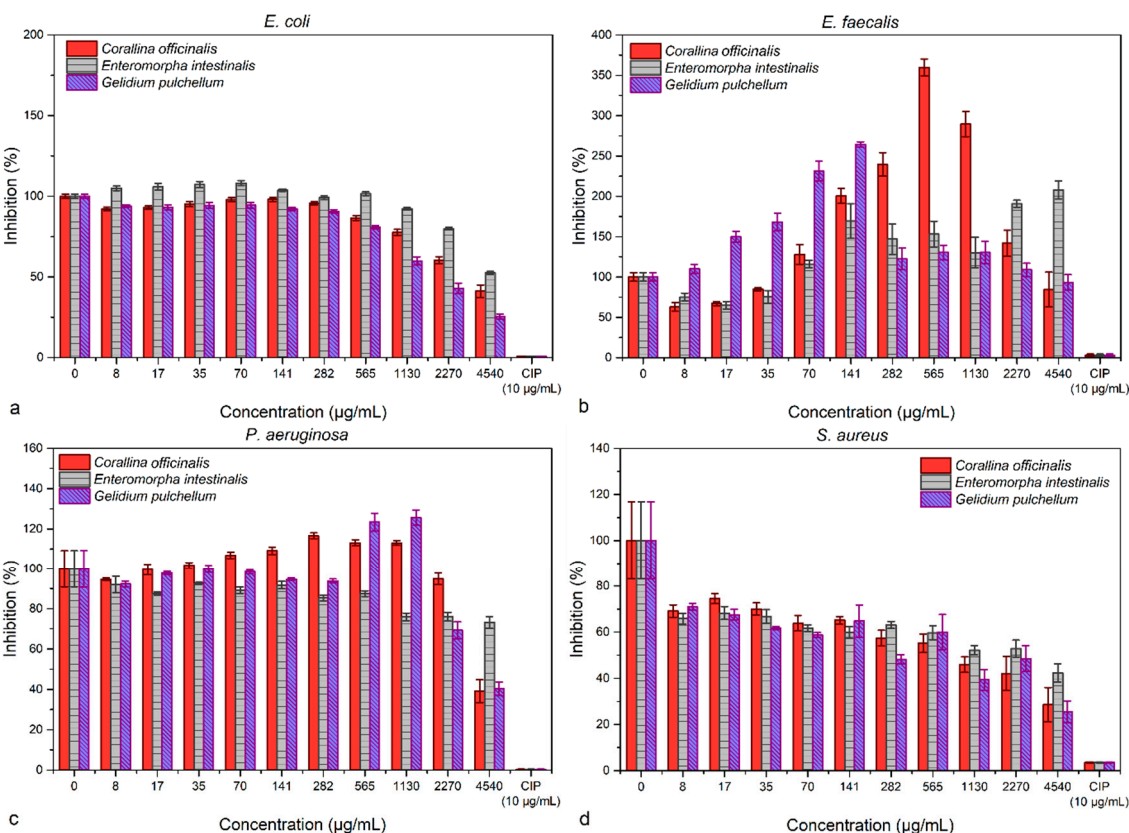

**Figure 4.** The antibacterial effects of the algae extracts assessed through the microdilution method against *E. coli* (**a**), *E. faecalis* (**b**), *P. aeruginosa* (**c**), and *S. aureus* (**d**).

### 3.4. Adsorption and Release Tests

The visual appearance of the pressed pellet obtained from crab shell powder loaded with algae extract is shown in Figure 5 (inset). The pellets are resistant to fracture and abrasion in the dry state. However, the pellets disintegrated in a few seconds upon immersion in water under stirring conditions.

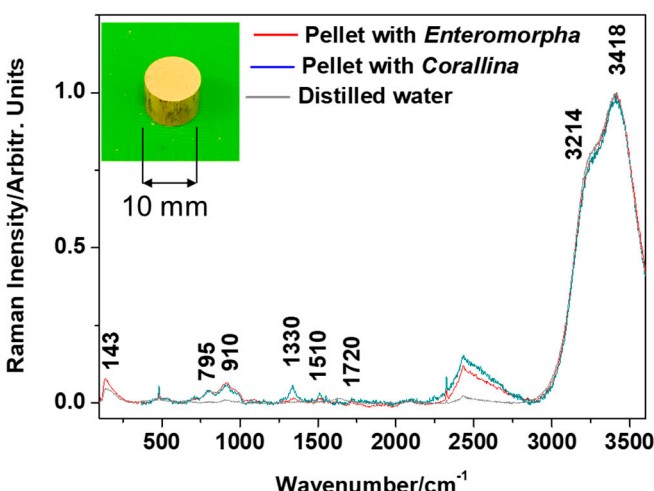

**Figure 5.** Raman spectra acquired from the water samples collected 1 h after the addition of respective single pellets made of crab shells and loaded with either the aqueous extract of *E. intestinalis* or *C. officinalis*.

Raman spectra acquired from subsamples of supernatant from either pellet dispersion vessel, taken in hourly intervals, reveal the presence of carotenoid signals ($\upsilon_1$ band at 1510 cm$^{-1}$, arising from the C=C stretching) and the bands of organic compounds, including C-H rocking and bending modes in the 700–1800 cm$^{-1}$ range (Figure 5). These bands are close to phenol bands at 803, 912, and 1330 cm$^{-1}$ observed in the SERS signal of algae extracts (Figure 5). The remaining bands, at 143, 3214, and 3418 cm$^{-1}$, arise from O-H vibrational modes of water used for the pellet's dispersion and as the colloidal AgNPs dispersion medium. The bands of organic compounds appeared at the first sampling (1 h) and remained without notable changes throughout further sampling intervals.

## 4. Discussion

The dissolved ions concentrations identified in the three algae species investigated in our paper were higher than the levels reported in the literature for the brown algae *Fucus vesiculosus* [22] and the seaweed *Ulva rigida* [34]. The same situation also applies to the seagrass *Posidonia oceanica*, green algae *Penicillus capitatus*, *Codium bursa*, *Ulva lactuca*, red algae *Asparagoposis* sp., or brown algae *Sargassum* sp. [35], except for the F$^-$ content. The detection of NO$_3^-$, PO$_4^{3-}$, and K$^+$ considered as main plant nutrients, and the additional content of Ca$^{2+}$, Mg$^{2+}$, SO$_4^{2-}$, and F$^-$ make the extracts of the three tested species an interesting liquid organic fertilizer.

UV/Vis and SERS data showed the highest intensity bands from *E. intestinalis* aqueous extract, suggesting that this sample is richer in phenolic compounds than the *C. officinalis* and *G. pulchellum* are. Phenolics have a wide range of biochemical functions, serving as pigments, antioxidants, signaling compounds, and a defensive mechanism [24]. The antioxidant activity of phenolics is mediated by their hydrogen donation, metal ions chelating, and low-density lipoproteins oxidation inhibition capabilities [24]. Crop plants generally benefit from rhizosphere treatment with phenolic compounds in terms of germination capacity, biomass, and growth in size [36], while specific benefits are exhibited by rice, for instance, where protocatechuic and vanillic acid treatment also promotes seedling survival during the submergence period [37].

The content of phenolic compounds presumably also accounts for the antibacterial properties observed in dispersed powdered algae biomass [38–42].

The heavy metal levels detected both in the starting dry algae biomass and in the solid extraction residual were significantly higher than those determined in the extracts (Supplementary Table S2). A low metal extraction yield from the leaves of the seagrass *Posidonia oceanica* was also noted by Castaldi and Melis et al. [43]; however, these researchers noted a higher metal extraction yield when using EDTA as a solvent. This points to the supposition that heavy metals in algae and marine plants may exist in a form that is not readily dissolved in water, hence advocating the aqueous extract over the raw biomass to avoid soil contamination. The leftover solid biomass may still be useful as a heat energy source, as previously suggested for other algae species [44] and the seagrass *Posidonia oceanica* [45].

Relevant to the environmental legislation, the levels of all heavy metals in the extracts of the three tested species are below the safety threshold prescribed for liquid biofertilizers by the European Union [46]. Furthermore, local legislation regarding the exploitation of particular algae species has to be consulted before planning a waste biomass processing facility; for instance, the biocenosis centered around *C. officinalis* is considered endangered in the south-eastern Adriatic; hence, its exploitation in nature is discouraged.

Moreover, Fe results determined in the investigated dry algae biomass were in the range reported by Taboada et al. [34] in *Ulva* sp., Shaw and Liu [47] in *Porphyra* sp., and García-Casal et al. [48] in *Sargassum* sp. or *Gracilariopsis* sp. *C. elongate*, and the *E. intestinalis* analyzed in the current study showed similar Mn, Zn, and Cu concentrations with those reported by Masoud et al. [35] for *Posidonia oceanica*, *Penicillus capitatus*, *Codium bursa*, *Ulva lactuca*, *Asparagoposis* sp., and *Sargassum* sp., but higher in the case of *G. pulchellum*.

Likewise, the Ni and Cd contents agree with the data previously reported in the literature on other algae species [35].

The Raman bands at 795, 910, 1330, 1510, and 1720 $cm^{-1}$ observed in the spectra of suspension subsamples of dissolved crab shells/algae extract pellets are caused through the dissolution of previously loaded nutrients (extract) onto the mineral carrier (shell powder) in respective pellets. This observation proves that the nutrients and biogenic compounds extracted from algae can easily be stored in a dry state by loading on a crab shell nanoporous biomineral matrix. Subsequent drying and pelleting of the shell/extract mixture enables convenient handling and packing of the resulting biofertilizer complex.

## 5. Conclusions

A series of cross-disciplinary tests were carried out in order to prospect the suitability of aqueous extracts of marine algae *Enteromorpha intestinalis*, *Corallina officinalis*, and *Gelidium pulchellum* biomass for use as a fertilizer, instead of landfilling as a common waste of biological origin. The extract exhibits favorable properties, such as nutrients content and a lower heavy metals content than the raw biomass and phenolics content, possibly linked to the mild antimicrobial activity. This makes the extracts an attractive and suitable solution for the handling of algae biomass build-up through nutrient recycling, while, at the same time, allowing a contaminant-free, nutrient-rich irrigation medium for greenhouse or outdoor crops. Furthermore, the algae extract can be combined with crab shell powder and pressed into pellets to allow dry storage and transport and easier handling.

**Supplementary Materials:** The following are available online at https://www.mdpi.com/article/10.3390/su13168777/s1, Table S1: Assignments of the SERS band observed in the spectra acquired from extracts of *Corallina officinalis*, *Gelidium pulchellum*, and *Enteromorpha intestinalis* to specific atomic vibrations of polyphenols [1] and polysaccharides [2], Table S2: Heavy metals content (mg $kg^{-1}$ dw) and extraction rate (%) of individual heavy metals under boiling at 100 °C in the presence of 2% sol. of $Na_2CO_3$ for 100 min (preserved in a separate file), Figure S1: Extraction rates of individual heavy metals determined in the fresh extracts after boiling at 100 °C in the presence of 2% sol. of $Na_2CO_3$ for 100 min, shown as fractions extracted (mg $kg^{-1}$ of liquid extract) from the dry algae biomass (starting material, mg $kg^{-1}$ of dry weight).

**Author Contributions:** Conceptualization, F.N. and I.-V.G.; methodology, F.N., I.-V.G., A.C. and R.H.; validation, F.N., I.-V.G., A.C., S.C.P. and I.M.M.; formal analysis, F.N., I.-V.G., A.C. and I.M.M.; investigation, F.N., I.-V.G., A.C. and S.C.P.; resources, S.T. and R.H.; data curation, F.N., I.-V.G., A.C. and I.M.M.; writing—original draft preparation, F.N., I.-V.G. and S.C.P.; writing—review and editing, F.N., S.C.P. and S.T.; supervision, F.N. and S.C.P.; project administration, F.N.; funding acquisition, F.N. and I.-V.G. All authors have read and agreed to the published version of the manuscript.

**Funding:** This study was funded by the Babeș-Bolyai University through the Grants for Young Researchers Programme (contract no. GTC-31992/02.06.2020) and the project: Entrepreneurship for innovation through doctoral and postdoctoral research, POCU/380/6/13/123886 co-financed by the European Social Fund, through the Operational Program for Human Capital 2014–2020. S.C.P. and R.H. acknowledge funding from the PN-III-P2-2.1-PED-2019-4777, Acronym: BlueBioSustain.

**Institutional Review Board Statement:** Not applicable.

**Informed Consent Statement:** Not applicable.

**Data Availability Statement:** The data that support the findings of this study are available within the manuscript, its supplementary information, and from the corresponding author upon reasonable request.

**Acknowledgments:** The authors wish to acknowledge Carmen Roba from the Faculty of Environmental Science and Engineering (Babeș-Bolyai University Cluj-Napoca) for the helpful technical assistance provided during the heavy metals analysis. F.N. and I.-V.G. acknowledge the POCU/380/6/13/123886 project co-financed by the European Social Fund through the Operational Program for Human Capital 2014–2020.



**Conflicts of Interest:** Iolanda-Veronica Ganea is an employee of MDPI; however, she did not work for the journal *Sustainability* at the time of submission and publication.

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
