# Peer review of "A New Biofertilizer Formulation with Enriched Nutrients Content from Wasted Algal Biomass Extracts Incorporated in Biogenic Powders"

_sustainability, doi:10.3390/su13168777_

Round 1

Reviewer 1 Report

The paper is concerned with an important issue of providing crop nutrients in a sustainable way. However, I have indicated a number of specific issues using 'sticky notes' in the attached file. I feel that a number of issues need expansion/more explanation:

  • Why does the algal material NEED to be removed from the coastal regions - surely in the sea it provides resources for marine organisms. If it is present to excess is this because of pollution - if so the surely this should be addressed first?
  • Why was the production of an aqueous extract examined? Why not just use the algal biomass as a solid fertiliser (perhaps after composting)?
  • How do the levels of major nutrient elements in the extracts compare with commercially available alternatives? Why was the particular extraction ratio (26:1) selected? The levels of nitrate and phosphate often seemed quite low.
  • Are the antimicrobial effects really a good thing? Might this not interfere with normal soil processes?

Author Response

The comments listed in the submission system are answered in a separate Word document attached.

The comments provided as "Sticky notes" to the original manuscript are answered in the second part of the respective manuscript..

Reviewer 2 Report

The paper conducted some characterizations of algal biomass but failed to establish the linkage that how these parameters can affect the performance of biofertilizers. 

  1. More introduction about the application of waste biomass to biofertilizers need to be discussed.
  2. For complicated compounds, UV-spectrum and Raman are not good ways to analyze the composition of materials.
  3. It is hard to conclude that the phenolic compounds are the cause of the antibacterial effect. Moreover, How these compounds affect the growth of plants.

Author Response

Please see the attachment. The comments were answered in the attached pdf FILE.

Round 2

Reviewer 2 Report

As all of my concerns are addressed, I recommend publishing the paper in the present form. 

Author Response

We thank the Reviewer for improving the quality of our article.